# Desacralization of Religious Concepts: The Prophecy from the Perspective of the Iranian Reformist Scholar Seddigha Wasmaghi

**Abbas Poya**

Department for Islamic-Religious Studies, FAU Erlangen-Nuremberg, 91052 Erlangen, Germany;
abbas.poya@fau.de

**Abstract:** This article examines, how the reformist attempts of some Iranian religious intellectuals—consciously or unconsciously—lead to the desacralization of Islamic concepts, using the Iranian jurist and activist Seddigha Wasmaghi as an example. The reformists are, as will be shown with reference to Wasmaghi, concerned with establishing that the normative as well as the theological assumptions in Islam are results of human cognition. Any idea that is qualified as a human assumption, i.e., not sacred and thus open to challenge, can be critically examined, re-read, and perhaps even changed or overruled. Such approaches include, for example, Mohammad Mojtahed Shabestari's understanding of the Qurʾan as a 'prophetic reading of the world' and ʾAbdolkarim Sorush's interpretation of revelation as 'prophet's dreams'. Among the most recent attempts of this kind is Seddigha Wasmaghi's perception of 'prophecy as a human construction'. This argument is presented and critically analyzed in this paper.

**Keywords:** desacralization; Islamic reformism; Iran; Shia; religious intellectualism

## 1. Introduction

The underlying thesis of this article is that the intellectual approaches of some Iranian reform thinkers are leading towards debunking the sacred in many Islamic conceptions, an endeavor that I would like to call a desacralization attempt. I argue that these attempts should be seen in close connection with experiences of Iranian reform thinkers with the Islamic rule practices in Iran. This is another thesis resonating in my reflections discussed here.

The sacred within a religion seemingly bothered hardly anyone in Iran, as long as religion had no political power and resided in the free spaces of individual life. However, as soon as it led politics, and with it regulated the public sphere, the sacred seemed to constitute a great barrier, preventing criticism of the ruling power. Therefore, I believe, some reform thinkers have begun to question the sacredness of Islamic ideas. The central concern of these intellectual efforts, to use a formulation of the German sociologist Hans Joas, is the questioning of the "power of the sacred" (Joas 2017). It can also be assumed that Wasmaghi pursues the same motive with her thesis that prophecy is a human construction.

It is important to note that desacralization does not equate to the disappearance of God or religion from society. Rather, it means cutting the ties of some phenomena and assumptions of God, while faith in God and the practice of religious rituals can still be nurtured. Accordingly, when Mohammad Mojtahed Shabestari maintains that the Qurʾan is the result of Muhammad's contemplation and interpretation of the world (Poya 2017), he, as a professing Muslim, hardly intends to question the religious in Muhammad's messages and the existence of God per se. Rather, his intention is to enable uninhibited reflection on the Islamic texts that have been passed down to us and understand them on the basis of contemporary human knowledge. Likewise, when ʾAbdolkarim Sorush, to cite another example, describes Muhammad as a 'power-seeking man' (*qodrat-talab/eqtedār-garā*)[1] in

a controversially debated webinar series entitled *Din wa-qodrat* ('Religion and Power'), who sought the establishment and expansion of his worldly power by all means, he too, as a devout Muslim, seeks not to question the existence of God or the spiritual aspects of Muhammad's messages. Instead, he tries to see Muhammad the human from a levelheaded historical and source-analytical perspective, as he fundamentally believes that both the Qur'an and Muhammad can be subject to critical reflection (Sorush 2015). The same is true for Seddigha Wasmaghi. On the one hand, she tries to consider Muhammad, like all other prophets, not as a prophet chosen by God but as a human being searching for God; on the other hand, she always affirms her belief in God and the meaningfulness of many religious concepts.

This article will analyze Wasmaqi's most relevant works and, in particular, her last book, The Way of Prophecy, using a text-analytical method, with the aim of locating Wasmaghi's thinking in today's religious discourse in Iran. Moreover, the article will highlight the specificity of her thesis that prophecy is a human institution.

## 2. Wasmaghi: A Short Introduction

In the West, there is little research on Wasmaghi and her ideas. The only comparatively detailed analysis was published in 2022 by Ali Akbar (Akbar 2022, p. 1046), who is associated with the National Centre for Contemporary Islamic Studies at the University of Melbourne. The article focuses on Wasmaghi's reformist approaches to Islamic norms, i.e., her earlier works, which I will briefly introduce later. That said, Akbar's article has been of great help to me, especially in outlining Wasmaghi's biography. However, no comprehensive analysis is currently available on Wasmaghi's thesis, discussed here, namely the aspect of prophecy as a human institution, to which she dedicates her latest book.

The first thing to note here is that the religious reform debates in Iran, despite their progressive claims, are mainly dominated by men. Very few women have been able to establish themselves as spokespersons in this field. Seddigha Wasmaghi, born in 1961, is an exception. Other women's names circulate in expert circles, such as A'zam Puyāzāda,[2] assistant professor at Tehran's Faculty of Theology, or Fātema Tawfiqi,[3] assistant professor at Qom's University of Religions and Denominations. None, however, has been able to achieve Wasmaghi's level of political-intellectual prominence or influence.

As I have explained in more detail elsewhere (Poya 2023, pp. 8–9, 12–14), many of today's reform approaches, by Shia in general and by Iranians in particular, can be traced back to the actors' negative and disappointed experiences with the ideology and practice of Islamic groups, including the Islamic Republic of Iran. Wasmaqi is no exception. As discussed later, Wasmaqi is a convinced Muslim woman, but obviously rejects the exclusivist reading of Islam and the resulting undemocratic politics. With her thesis that prophecy should not be understood divinely, but humanly, she tries, alongside her pioneers Shabestari and Sorush, to pave the way for a pluralistic society.

By education, Wasmaghi is a legal scholar, but she has also excelled in other fields. She is a poet, reformist politician, and theologian. She studied at a women's seminary for three years in the 1980s, following which she obtained a Bachelor of Arts degree from the Faculty of Theology at Tehran University. Wasmaghi continued her studies at the same university, graduating with a doctorate in Islamic jurisprudence. After receiving her doctorate, she became a member of the theology faculty in 1991. Wasmaghi became particularly prominent in the political sphere after the election of reformist Mohammad Khatami as president in 1997; she served as a member of the Tehran City Council during his administration from 1999 to 2003. In response to President Ahmadinejad's reelection in 2009 and the emergence of the Green Movement, she joined the protest movements in Iran. However, like several other reformists, she ultimately decided to leave the country and go abroad. In 2011, she was appointed a visiting professor at the University of Göttingen in Germany and later at Uppsala University in Sweden, where she remained until her return to Iran in 2017. Upon her return, Wasmaghi was initially interrogated but eventually released. Recently, she was summoned to appear before Iran's Revolutionary Court but

refused to attend (Akbar 2022, p. 1046). Notably, after the Zan Zendagi Āzadi Movement (2022), she, like many other activists, began calling for more than just social and legal reforms of the current constitution. In one of her latest articles, "No to the Vulgarity of Oppression" (*Na ba ebtedhāl-e sarkubgari*), she advocates a referendum allowing Iranian citizens to decide whether they still want the Islamic Republic.[4]

When analyzing the reform discourse in Iran, it must be mentioned that the disparities between religious reform thinkers have increased. In the 1980s and 1990s, theorists such as Sorush and Shabestari were regarded as reliable providers of ideas for many of their followers and their thoughts as solid sources of inspiration; however, some of their former supporters, such as Mohsen Kadivar (Kadivar 2022), Abu l-Qāsem Fanāi (Fanāi 2022), and Arash Naraghi (Naraghi 2008), are now distancing themselves from them. They accuse them of overstepping boundaries with their reformist ideas and questioning the foundations of Islam on some issues. Central to their differences is that Sorush, Shabestari, and Wasmaghi deem it important and necessary to question every Islamic or Shiite concept, be it revelation, prophecy, or the infallibility of the Shiite imams, an approach that leads to a general desacralization of the Islamic religion. Other reform thinkers, such as Kadivar, Naraghi, and Fanāi, neither want to go that far nor see any need for it, at least not under the present circumstances. Thus, the latter even try to distance themselves from the former by name, calling themselves 'new-thinkers' (*nawandish*) and the others 'revisionists' (*tajdidnazar-talab*) (Kadivar 2022). They sometimes implicitly express their skepticism as to whether the former truly still believe in Islam, and question whether in claiming to be Muslims they are not actually engaging in *taqiyya*, i.e., concealing their true beliefs cf. (Kadivar 2022). One of the strongest critiques of the earlier reform thinkers, especially Sorush, was recently published by his former student Akbar Ganji.[5] He tackles Sorush's ideas about God and describes his reasoning as a confused, incoherent mess. He notes, among other things, that Sorush's conception of God leads to the rejection of revelation and prophecy and, ultimately, reduces God to a mere name (Ganji 2022, pp. 1–2, 29).

Following the publication of her most recent book *Masir-e payāmbari* ('The Path of Prophethood') in 2021, Wasmaghi has been counted among the Sorush and Shabestari ranks (Kadivar 2022).

## 3. Wasmaghi's Thought

In her academic work, Wasmaghi primarily addresses Sharia law issues with a special focus on women's rights. For instance, in her book *Zan, feqh, eslām* ('Women, Jurisprudence, Islam'), published in Iran in 2008 and available in English, she criticizes the particularly prevalent portrayal of women in Islamic jurisprudence. She argues that while other Islamic disciplines such as philosophy, theology, and mysticism also portray an unequal representation of women, these sciences are not normatively relevant. It is jurisprudence that defines norms and puts women at a disadvantage in ritual practice, family law, and pursuance of certain professions (Wasmaghi 2008, pp. 13–16). Wasmaghi then attempts to rectify the legal inequality between men and women by contextualizing traditional norms and critically reflecting on hadiths and qur'anic verses on which the traditional stance regarding this issue is based cf. (Wasmaghi 2008, pp. 17–20, 37–42, 75–81, 113–26). In doing so, she still remains largely faithful to traditional notions of revelation and prophecy, or at least does not question them at any point.

In 2009, Wasmaghi attempted to publish her second critical book on Sharia, *Bedhā at-e feqh wa-gostara-ye nofudh-e foqahā* ('The Capacity of Jurisprudence and the Sphere of Jurists' Influence') in Iran, but was prevented from doing so. It was not until 2013 that she was able to publish it online in Uppsala. Therein, she follows a methodology very similar to that of her previous work. She reiterates here that Islam is not to be equated with the traditional Sharia understanding of it, and that Islamic norms can indeed be modified and interpreted in a contemporary manner using historical, theological, and exegetical approaches. Theologically, she continues to conform to the traditional ideas on revelation and prophecy (Wasmaghi 2013).

In 2017, Wasmaghi published her next book online in Uppsala, *Bāzkhāni-ye shariʿat* ('Rereading of the Sharia'), which once again takes Islamic jurisprudence as its subject. Therein she reflects critically on the question, why today everything Islamic is reduced to Islamic jurisprudence. She claims that other Islamic sciences, such as theology, legal theory, and history, which are more rationally oriented and simultaneously serve as the basic sciences for jurisprudence, are increasingly neglected, resulting in jurisprudence becoming more and more literalistic and dogmatic (Wasmaghi 2017, pp. 7–8). Moreover, she tries to justify, with historical, theological, and hermeneutical arguments, that Islam or the Sharia cannot be equated with Islamic jurisprudence. At the same time, she remains predominantly faithful to the traditional notions of revelation and prophecy. Accordingly, she uses the traditional terms such as the 'sending' (*beʿthat*) of the Prophet cf. (Wasmaghi 2017, p. 272) and the 'sending down' (*nozul*) of qurʾanic verses[6]. At one point, she also explicitly affirms that there is no doubt that the qurʾanic verses were revealed to the prophet and he recited them to his followers.[7]

Nevertheless, in her latest book, *Masir-e payāmbari*, published online in 2021, Wasmaghi posits a thesis that is theologically, and arguably juridically, momentous, namely that prophecy is something earthly or human.

Looking at the chronology of her works and the ideas discussed therein, it seems as if Wasmaghi has identified a certain incoherence in her previous reform ideas. Whereas in her previous works, like many other reform thinkers, she still omits the theological assumptions and only attempts to discuss new reformist considerations in the area of legal issues, in her latest book she practically turns the traditional theological conception upside down. Evidently, she ought to have concluded, according to my speculative explanation of this radical leap, that consistent reform of normative matters—such as equal rights for women, rejection of *ḥadd* punishments and hijab coercion, recognition of freedom of expression, as well as efforts to establish a human rights-oriented political order—cannot take place without reconsidering theological assumptions. After all, if the Qurʾan is considered to be God's and was revealed through God's chosen Prophet Muhammad, the regulations from the Qurʾan and Hadith would have to apply to all times and places. However, many passages in these texts are contrary to the reform ideas just mentioned. Islamic scholars have tried to adapt the Islamic norms to the new temporal and spatial circumstances, to a certain extent, by means of traditional Sharia law. However, they have not succeeded in fully harmonizing these norms with contemporary values.

Therefore, in her latest work, Wasmaghi attempts to question one of the fundamental assumptions of Islamic theology—prophecy as a sacred institution—and thereby pave the way for a purely rational approach to the Qurʾan and its statements. She thus joins Shabestari and Sorush in the ranks of thinkers who must contend not only with traditional scholars, but now also with their former colleagues and fellow combatants.

In her book, Wasmaghi tries to provide a general theoretical basis for her reform ideas by desacralizing the prophet. She thereby challenges the traditional concept of prophecy and argues that it is not God but a human being, i.e., ultimately Muhammad himself, who established the institution of prophecy. A prophet, she argues, is a human being who develops religious teachings by means of his own reflections and self-reflection and transmits them to others so that they may find bliss (*saʿādat*) and salvation (*rastgāri*). With these two terms, Wasmaghi certainly includes worldly happiness and well-being. Thus, if prophecy and religious teachings are of worldly or human nature then any rethinking and renewal in religion can be understood as a legitimate process (Wasmaghi 2021, pp. 12–13).

## 4. Prophecy as a Human Institution

According to the traditional concept, belief in the one God (*tawhid*) is the first and most significant pillar of Islam. If, however, the sequence of religious knowledge is observed in its emergence, the institution of prophecy represents the heart of faith, at least in the monotheistic religions, including Islam. After all, religious teachings, norms, and rites, as well as faith in God, are first taught to people through a prophet. Therefore, the prophet

is the linchpin of monotheistic religions. Islamic scholars traditionally believe that God chose prophets to bring His messages to people and lead them to bliss and salvation. Nevertheless, this view was also conveyed through the prophets themselves.[8]

In the traditional Islamic understanding, the assumption that the prophet was chosen by God and conveyed his message to people establishes the sacredness and inviolability of religious assumptions and norms. This is why a traditional scholar's scope for adapting Islamic precepts to changing circumstances is also extremely limited in those cases that are non-definitive (*zanni*), or, in other words, do not establish Islamic or confessional identity. What is definitive or identity-forming (*qatʿi*)[9] for Islam or the respective Islamic denomination, such as many ideas traditionally held about God, prophecy, and revelation, remains sacred and inviolable in all cases.

By focusing on the institution of prophecy, Wasmaghi sets the belief concepts in Islam in a traceable order with regard to the chronology of religious knowledge. Moreover, by considering this institution as a category created by a human being or the Prophet, she pioneers a way for rethinking all aspects of religion, including traditional ideas about God, creation, and revelation. This is the central statement of her book, likewise entitled 'The Way of Prophecy':

> "For thousands of years, human beings have believed that God chooses the prophet. But perhaps God was chosen by human beings. For centuries, human beings have believed that the path of prophecy runs from God to human being. But perhaps the path of prophecy goes from human being to God". (Wasmaghi 2021, p. 13)

Seen from this perspective, it is not God who, out of responsibility, grace, and benevolence, has chosen prophets and entrusted them with the task of conveying His message to people. Rather, it is the people or the prophets who have searched for God, established a relationship with Him, communicated with Him, and in this way formulated teachings for other people. As such, according to Wasmaghi, the relationship between human being and God does not go "from heaven to earth, but from earth to heaven" (Wasmaghi 2021, p. 19).

In keeping with Wasmaghi's thesis, revelation (*wahy*) is the result of the Prophet's own reflections (Wasmaghi 2021, p. 141). As such, *wahy* is not something that God reveals to the prophet or communicates to him through an angel. Rather, God and the contents of revelation are what the human being discovers in his search or what reveals itself to him through 'uncovering' (*mokāshafa*). Thus, everything that Muhammad proclaimed and reported about God, His attributes, and the creation was the result of his own perceptions, which were expressed in the Qurʾan (Wasmaghi 2021, p. 142).

Therefore, if the prophet is not a person chosen by God for a special task, then he is an ordinary person who can also make mistakes and be subject to criticism:

> "It is a tangible fact that Jesus was a human being. To make him the incarnate God means abandoning the realm of facts and entering the realm of imagination and fantasy. [...] Muhammad, the Prophet of Islam, was also a human being like all other human beings. [...] Some have then considered him sinless and many others even faultless. However, these are assumptions that are not even consistent with the statements in the Qurʾan itself". (Wasmaghi 2021, p. 11)

Wasmaghi alludes here to the passages in the Qurʾan that describe Muhammad as a human being like all others: "Say: I am only a human being like you [...]." (Q 18:110, 41:6)

In her view of prophecy, Wasmaghi removes the basis for the traditional conviction that Islamic teachings come directly from God and are therefore supra-temporal. By doing so, she opens the way for her reflections on traditional religious beliefs, which are based on human reason and experience.

## 5. Wasmaghi's Line of Reasoning

Wasmaghi's argumentation starts with a question that many religious reform thinkers in Iran might be asking themselves, especially because of their disappointment with the

Islamic Republic, to whose founding they had often actively contributed and in which they had initially placed great hopes: Why is it that Sharia, which, according to its own claim, represents the will of God and is supposed to provide happiness and well-being for people, has failed to do so, in Iran or elsewhere in the world? The "simplest, most transparent, and most meaningful answer" to this question, according to Wasmaghi, is that "Sharia in the sense of divine laws does not exist" (Wasmaghi 2021, p. 12). With this radical statement, she does not seek to deny that religious norms can also cause well-being and happiness, but rather that they must not be viewed dogmatically and need to be adapted to the respective circumstances by virtue of reason and experience (Wasmaghi 2021, pp. 11–12). In doing so, she pleads for a "merciless" (*bi-rahmāna*) rational-critical examination of Islamic ideas and beliefs, "refraining from [any] interests" (*ghayr-e maslahat andishāna*) (Wasmaghi 2021, p. 13).

As briefly mentioned above, Wasmaghi had already fundamentally questioned the traditional ideas concerning the Sharia and many Islamic norms, including the position of women, in her earlier publications. She argued that the accepted Sharia legal norms ultimately reflected the views of traditional scholars. Furthermore, she pleaded for Sharia to govern solely moral matters and acts of worship. Other issues, such as civil and criminal law, should be negotiated outside of Sharia law.[10] With her latest book and its thesis that prophecy is a human construction, she wants to take it a step further and reiterate that Sharia law, indeed fundamentally all Islamic teachings, are the result of the Prophet's reflections and interactions with God.

Clearly, Wasmaghi cannot justify such a thesis, which is articulated against traditional theological ideas within the framework of the theological logic of argumentation that has developed historically in Islam. Instead, Wasmaghi challenges this logic and attempts to examine her religion from the outside, using general scientific and historical findings (Wasmaghi 2021, pp. 163–65).

According to Wasmaghi, if one assumes, like the traditional scholars, that God elected prophets out of mercy (*rahma*) and benevolence (*lotf*) and sent them to people to guide them on the right path, then prophets must have existed in all eras and in all parts of the world, because God's mercy and benevolence encompasses all people. Moreover, the teachings of all prophets would then have to be identical, as it would contradict God's mercy and goodness if he were to differentiate between people in conveying his message. Nevertheless, Wasmaghi argues that the historical facts and the ongoing religious and theological disputes between people indicate the opposite cf. (Wasmaghi 2021, pp. 13–14, 18–20, 73, 101–3), for all the prophets narrated by the Abrahamic religions would have lived exclusively in the time of the agrarian societies.[11]

The principle of benevolence (*lotf*)[12] plays an important role in classical Muʿtazilite[13] and Shiʿite[14] theology and essentially states that since God is perfect and merciful, he must guide people on the right path. Therefore, he appointed prophets and sent them to the people. This assumption underlies many other theological questions as well, such as why religious precepts are obligatory and why, as is believed by the Shia, it is necessary that there be imams. Wasmaghi, however, sees this principle as contradictory to the historical reality. For if God, by virtue of his goodness, was obligated to send prophets endowed with salvific teachings to mankind, then why were there no prophets sent to mankind before the advent of agrarian societies, and why were later prophets sent only to certain regions, leaving people in other regions without prophets? (Wasmaghi 2021, p. 103).

In order to address these questions, Wasmaghi proposes an entirely different understanding of prophecy. In reality, creative human beings or prophets sought God, chose him, and established a relationship with him. In Islam, the result of this relationship was the message that the Prophet Muhammad conveyed to people, and only then did he become a prophet (Wasmaghi 2021, p. 19).

## 6. The Implications of Wasmaghi's Thesis

If one considers prophecy and, with it, revelation, in the way Wasmaghi suggests, as human phenomena that are at the same time connected to God, one has the theoretical basis

for scrutinizing Islamic scholarly disciplines, be it theology, Qurʾan and Hadith sciences, or jurisprudence. Wasmaghi gives some examples of how far-reaching the implications of her thesis can be.

For instance, she classifies the creation story in the Qurʾan, according to which human beings were created from earth (3:59), dry matter (15:26), or clay (23:12), on the basis of today's biological and archeological findings as a myth on which the earlier people relied and which today is to be deemed obsolete (Wasmaghi 2021, pp. 25–26). Furthermore, she regards the Qurʾanic report, in which, according to her interpretation, the creation of the heavens and the earth was carried out in eight days in one passage (41:9–12) and in six days in another (25:59), as contradictory. Concurrently, she states that the ideas in the Qurʾan regarding this, as well as all other explanations of creation in pre-modern religions and cultures, were attempts undertaken outside of science and later challenged by it (Wasmaghi 2021, pp. 26–30).

Wasmaghi also contests the commonly accepted assumption in traditional Islamic theology that monotheism practically existed with the first human being on earth. For her, belief in a single God is a statement of faith at the end of a complex developmental process in which human beings tried to find answers to essential questions. Historical and archaeological findings revealed that on their long quest, human beings experienced a wide variety of beliefs, from totemism and shamanism to idolatry and polytheism, each in its respective form, until they were convinced of monotheism. As such, the assertion that monotheism existed from the very beginning of humanity's creation is an unfounded and scientifically unsustainable one (Wasmaghi 2021, pp. 31–41). Moreover, all monotheistic prophets known to humankind stem from the time of agrarian societies. Therefore, people before that time could not have known about monotheism (Wasmaghi 2021, p. 73).

Another dimension of Wasmaghi's thesis is her view that God, like all other beings, and at times even human beings, is an object of development. Here, I believe, she essentially adopts the thesis of the well-known mystic Mohyi d-Din Ebn ʿArabi (d. 1240) of *wahdat al-wojud* ('oneness of being' in the sense of oneness of God and universe), even though she does not explicitly acknowledge it. For example, Wasmaghi says at one point: "The change and the development of the world are indicative of the change and the development of its creator. As the world expands and develops, so does its creator" (Wasmaghi 2021, p. 63). Furthermore, she states that "[i]t is through new and continuing creations that God learns about Himself, and with each experience He advances in his infinite path of evolvement" (Wasmaghi 2021, pp. 63–64).

A further outcome of Wasmaghi's thesis is evident in the field of Sharia legal norms, the author of which she considers to be human beings rather than God. She followed this view, as already mentioned, in her earlier works on law and Sharia. Here, however, she creates a theoretical foundation for her thesis by arguing that God does not fundamentally proclaim laws, but human beings themselves discern them: "The regulations and laws were not principally proclaimed by God; and it would not have been reasonable or just for Him to have proclaimed them" (Wasmaghi 2021, p. 106).

This view not only depicts Sharia as a human work, but it also contradicts the traditional position in theology held by the Muʿtazilites and the Shiʿites. The latter, with reference to the principle *qobh-e ʿeqāb be-lā bayān* ('punishment without explanation would be unjust'), generally maintain that people are informed of God's precepts by the prophets and can therefore be punished if they violate them. This principle is invalidated if one, like Wasmaghi, assumes that prophets chosen and commissioned by God did not in fact exist. This leads to a crucial question: By what standard will people be rewarded or punished in the hereafter, if God has not sent prophets to them, so therefore, they cannot know what is good and what is bad? Wasmaghi answers in her rational, human-oriented way that one's own intellect is the standard. Every human being recognizes good and evil according to their own intellect. If one follows one's own understanding, one will be rewarded; if one contradicts it, one will be punished (Wasmaghi 2021, pp. 105–6).

An additional aspect of Wasmaghi's thesis is her pragmatic interpretation of the goals of prophecy. For her, Muhammad's primary concern, as that of all other prophets, was not to impose regulations but to design a creative idea to solve the social issues at hand. His creative idea, states Wasmaghi, was the recognition of the One God: "In a hostile, polytheistic society permeated by war and bloodshed, Muhammad's proclamation of the unifying idea of faith in God helped the tribes to achieve unity and strength" (Wasmaghi 2021, p. 148).

Nevertheless, Wasmaghi does not believe that the fact that religion was concerned with solving social problems at the time of its origin implies that it should continue to regulate people's social affairs today. On the contrary, she considers religion and religiosity today to be a private matter. Questions concerning legal and social issues would have to be discussed outside the religious sphere (Wasmaghi 2021, p. 150).

In this way, Wasmaghi believes to be in a position to overrule many Islamic traditional norms that contradict today's generally accepted values, such as those of human dignity and individual freedoms. Thus, for example, the unequal legal position of women in the traditional jurisprudence can be rejected, as well as slavery and the *hadd* punishments (Wasmaghi 2021, p. 153). Since people themselves acknowledge that God requires from them to do what is good and refrain from doing what is evil, they themselves formulate the appropriate ethical and legal norms, depending on the spatial and temporal conditions in which they live.[15]

### 7. Some Reflections on Wasmaghi's Thesis

Since people themselves acknowledge that God requires them to do what is good and refrain from doing what is evil, they formulate the appropriate ethical and legal norms on their own, depending on the spatial and temporal conditions in which they live. That traditional scholars are not sympathetic to Wasmaghi's thesis is obvious. Her views, however, have also been severely criticized by some supporters of Islamic reformism. On the one hand, they contest her central thesis that prophecy proceeds from the bottom upward. On the other hand, they indicate inadequacies and contradictions in her argumentation.[16]

Yāser Mir Dāmādi argues that Wasmaghi's argumentation is fragmentary, for she only discusses the one opposing view, that the path of prophecy is from heaven to earth, in substantiating her thesis that the path of prophecy is from earth to heaven. Another alternative, for instance, that prophecy could represent a dialogue between earth and heaven, is not considered at all (Dāmādi 2022). Moreover, Wasmaghi claims that she approaches the question of prophecy without presuppositions. In doing so, she premises her argumentation on a certain degree of agnosticism and thus fundamentally excludes a metaphysical path to knowledge cf. (Dāmādi 2022).

Another of Wasmaghi's critics, Mahdi Yazdāni, points out that Wasmaghi's thesis is based, among other arguments, on the fact that archaeology has so far found no evidence of monotheism in the period before agrarian societies. He disagrees, stating that the absence of archaeological evidence for monotheism in that period is not an argument against its existence. Basically, Yazdāni continues, archeology cannot find any evidence for monotheism, as it categorically rejects the worship of material things (Yazdāni 2022). Furthermore, Yazdāni identifies contradictions in some passages of Wasmaghi's book. On the one hand, Wasmaghi tries to argue rationally for her thesis with great vehemence. On the other hand, she explicitly affirms her belief in the afterlife, for example, though this can hardly be justified rationally (Yazdāni 2022).

However, these are not the only questions or discrepancies readers may notice in Wasmaghi's work.

For instance, Wasmaghi is not always consistent in her argumentation in favor of a pluralistic understanding of religion, which is actually the basic tenor of her book and political stance. By 'grounding' or humanizing prophecy, Wasmaghi paves the way for thinking and speaking freely about religion and religious content. Wasmaghi's approach of viewing religiosity as a geographically and historically unevenly developing process is also

to be understood in a pluralistic direction, as well as showing the role of social circumstances in different religious imprints. In this sense, her openness to the different existing images of God in Abrahamic and non-Abrahamic religions is also to be interpreted (Wasmaghi 2021, p. 58). At the same time, however, Wasmaghi sees religiosity as a developmental process, i.e., as a movement progressing toward improvement, in which people eventually arrive at monotheism, that, consistently thought through, leads to evaluating beliefs according to their degree of development, which is more in agreement with an inclusivist and, strictly speaking, even exclusivist stance. This seemingly exclusivist attitude fits her remarks elsewhere, in which she rejects some religious views as "not correct" (*nā-dorost*). For example, she describes the divine nature of Jesus or the incarnation of God as a false, unfounded, and unreasonable doctrine (Wasmaghi 2021, p. 53).

Furthermore, Wasmaghi's argumentation is not always coherent. For instance, she posits that Muhammad was not chosen or commissioned by God, instead he sought God, established a relationship with Him, and then communicated the ideas that arose from that relation to people. Nevertheless, she leaves no doubt that she considers Muhammad a trustworthy person. At the same time, though, the Qurʾan affirms several times that God sent prophets with clear goals: that they warn people (71:1), that they urge them to worship God (23:23), and that people advocate justice (57:25). In verse 12:3, revelation is also explicitly described as a process that emanates from God: "We narrate unto thee the best of narratives in that we have inspired in thee this Qurʾan, though aforetime thou wast of the heedless."

Here, Wasmaghi is to be asked how Muhammad can report that God sent prophets and revealed the Qurʾan to him, when in Wasmaghi's view he was not commissioned by God nor received any revelation from Him at all.

Noteworthy is that Wasmaghi responds to Sorush's thesis that the Qurʾan includes dreams of Muhammad by asking why Muhammad never reported that the Qurʾan expresses his dreams. Still, the same question could be asked in relation to her own thesis: Why, then, the prophet never communicated that he was not appointed by God, even though there are many clear statements in the Qurʾan about God sending prophets to people with specific intentions (Wasmaghi 2021, p. 122).

Further criticism of Wasmaghi's study stems from her failure to situate her thoughts within the ongoing reform debates and their diverse currents. She references Shabestari's and Sorush's approaches to understanding the Qurʾan as Muhammad's work or his dreams at some points (Wasmaghi 2021, pp. 119–20, 140–41); nonetheless, she attempts to distinguish her own thesis from theirs. Thereby, all three essentially pursue the same idea of humanizing prophecy and desacralizing the Qurʾan. The same criticism is found in Ali Akbar's article on Wasmaghi's reformist reflections regarding Islamic law. Akbar notes many similarities between Wasmaghi's approaches and those of Shabestari, Sorush, Arkun, and Neuwirth. However, he finds no references to them in her works and thus states: "In the absence of clear reference to these writings, the extent to which Wasmaghi has been influenced by contemporary Iranian reformist thinkers or Western scholars remains unclear" (Akbar 2022, pp. 1046, 1061).

## 8. Conclusions

Wasmaghi's main thesis is that the path of prophecy is from the bottom up, from earth to heaven, and from human being to God. A human being, or the prophet, has created the institution of prophethood himself, making religion a human phenomenon. On this basis she does not justify her claim using established theological arguments, which in any case would not be possible, at least not from within the Islamic tradition itself. Rather, she discusses the question outside of religious presuppositions using historical, archaeological, and interdenominational findings.

Such an attempt aims at the desacralization of religious concepts and, here, specifically of prophecy, even though Wasmaghi does not speak of it. Nevertheless, she laments religious taboos in the introduction and reaffirms her strictly rational approach (Wasmaghi

2021, pp. 12–13), which clearly indicates the thrust of her work. Although Wasmaghi does not view her thesis as contradictory to her faith, she is aware of the danger of being misunderstood and denounced as an unbeliever. Therefore, she repeatedly emphasizes her personal faith as well as the fact that her thesis should not be understood in terms of Feuerbach's projection theory (Wasmaghi 2021, p. 167).

In doing so, Wasmaghi validates the thesis that desacralization does not automatically mean 'de-religionization' or atheism. Rather, attempts at desacralization seek to enable open reflection and discussion of Islamic ideas, regardless of category, without the fear of being ostracized from Islam. This goal is also evident in many reform-oriented figures such as Sorush, Shabestari, Kadivar, and Wasmaghi, despite their differing views.

The thesis that prophecy is a human construct relates closely to other assumptions that Wasmaghi presents in her latest book. Among them are the following: (a) Monotheism has not existed since the beginning of time; rather, it emerged in a later phase of human history. (b) The monotheistic prophets were shaped by the personal and social circumstances; therefore, their messages are connected to their respective traditions, cultures, and societies. (c) Religions contain general and timeless teachings; however, with regard to questions such as that of Being, God, the creation of human beings, and the way to salvation, they do not convey definitive knowledge that could not be questioned or rejected because their divine origin would be certain. (d) The messages transmitted by the monotheistic prophets or attributed to them include teachings that could be perceived today as unjust and unethical. (e) The traditional belief that God shows people the right path out of responsibility and goodness cannot be true, because otherwise, given the numerous differences between religions as well as within religions, God would have had to show the path of salvation and bliss much more clearly a long time ago cf. (Wasmaghi 2021, pp. 165–67).

With her effort to desacralize the Prophet Muhammad and the Qurʾan, Wasmaghi joins a long line of thinkers in Iran who have not only emerged today in the form of Sorush and Shabestari, but can also be observed from time to time in earlier times. Abu Bakr Mohammad bin Zakareyyā ar-Rāzi (d. 925),[17] to name but one example, also sought a comprehensive rational approach, even to theological questions. Among other things, he questioned the traditional perception of prophecy, arguing that it was contrary to God's wisdom and mercy to send prophets only to a certain people and exclude others (Amāni 2007). However, ar-Rāzi was criticized by Islamic scholars, such as Avicenna who stated that ar-Razi spoke out of ignorance; more modern take on ar-Razi is by Peter Adamson who argued that ar-Razi did not reject all religions (as was asserted by Isma'ilis), rather rejected the premise of using miracles to prove Muhammad's prophecy (Adamson 2021).

Regardless of whether and to what extent Wasmaghi's thesis will be accepted, it is characterized by its simple and direct way of tackling the issue. The central question underlying all religious-intellectual efforts by contemporary Iranian reform thinkers, including Wasmaghi's thesis discussed here, is, in short: How can a devout Muslim live in harmony with contemporary values? While many others, including Wasmaghi in earlier writings, try to solve only individual questions with the help of complex Sharia law and other traditional methods, here she proposes a universal and straightforward solution: God Himself does not decree any regulations; the believing person, instead, has always—and thus also today—been able to discern by himself and by virtue of his own reason what is good and what is evil, and to find his own way to happiness and a peaceful coexistence with others (Wasmaghi 2021, p. 161).

**Funding:** This research received no external funding.

**Conflicts of Interest:** The author declares no conflict of interest.

## Notes

[1] The series aired from 2019 to 2021 and is available on YouTube as well as his own website at drsoroush.com/fa (accessed on 22 August 2022).

[2] Cf. https://rtis2.ut.ac.ir/cv/puyazade (accessed on 19 November 2023).

[3] Cf. https://urd.ac.ir/fa/10938 (accessed on 15 September 2023).

[4] Cf. Wasmaghi, *Na ba ibtidhāl-e sarkubgari* (25 May 2023), online: www.zeitoons.com/111065 (accessed on 5 June 2023).

[5] Cf. on Akbar Ganji, see (Poya 2014).

[6] Cf. (ibid., p. 277).

[7] Cf. (ibid., p. 77).

[8] Wasmaghi also points this out in several places; cf. (a.o. Wasmaghi 2021, pp. 18, 163).

[9] On the categories 'definitive' (*qatʿi*) and 'non-definitive' (*zanni*) of legal theory cf. (Poya 2003, pp. 126–27).

[10] Her general approach in Wasmaghi, *Zan, feqh, eslām*. For a content summary by the author, see Rampoldi, "Interview with Sedigheh Wasmaghi".

[11] Cf. (ibid., pp. 20–22, 103).

[12] Cf. (ibid., pp. 101–3).

[13] On *lotf* by the Muʿtazilites cf. (Abrahamov 1993, pp. 41–58).

[14] On *lotf* by the Shia cf. (Sander 1994, pp. 168–69).

[15] Cf. (Wasmaghi 2021, p. 163). A detailed study of the question of good and bad behavior in Islam is provided in (Cook 2010).

[16] Two critical contributions should be mentioned here: (Yazdāni 2022; Dāmādi 2022).

[17] On ar-Rāzī cf. (Ulrich Rudolph, Islamische Philosophie: Von den Anfängen bis zur Gegenwart (Munich: C. H. Beck 2004), pp. 22–28).

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
