# Peer review of "Desacralization of Religious Concepts: The Prophecy from the Perspective of the Iranian Reformist Scholar Seddigha Wasmaghi"

_religions, doi:10.3390/rel14121452_

Round 1

Reviewer 1 Report

Comments and Suggestions for Authors

The article is well written, exposed and the main argument developed within it is clearly of high impact in the field of Islamic contemporary thought and Iranian studies.  

In parallel, the analytical spectrum is a topic of relevance in relation to the debate on the correlation between Prophecy, prophets and Monotheism in contemporary.

The only aspect which could be improved in the text is related to the Wasmaghi Islamic inspirational background in assuming a so clear position on Prophecy; since the early Kalam, mutakallimun's debate on the topic, about the bottom-up relationship between human being, prophetism and wahy has been developed in Islam. The issue that a Prophet is chosen by human beings and not by God is logical because none believer has never witnessed a proper dialogue between a human being and the divine. 

So, the author could better argue this argument in relation to Wasmaghi's cognitive roots to support such a interesting point of view.

Author Response

Dear reviewer,

Best regards

Reviewer 2 Report

Comments and Suggestions for Authors

I think the author could develop more on he historical conditions which help us to better understand Wasmaghi's thought. What were the main characteristics of this period? How these conditions oriented the Wasmaghi's academic works? The author could describe these conditions in line with the thoughts of Wasmaghi.

Comments on the Quality of English Language

The quality of English language is well.

Reviewer 3 Report

Comments and Suggestions for Authors

The article lacks an explanation of the research designs and methods. The author needs to describe the research methodology with a discussion why this methodology is the best approach for this paper. The article is more or less a summary of Wasmaghi's works. Further analysis is needed. The author is not clear about they are trying to accomplish in this article. What is the goal? Also, a literature review section is needed. This will help the author situate their work in the larger field, on the one hand, and demonstrate how this article departs from and builds on previous research, on the other. In other words, how does this article contribute to Islamic and religious studies? 

Round 2

Reviewer 3 Report

Comments and Suggestions for Authors

The article presents an original topic, which is well-analyzed and well-researched.